# Downtaper on Multimode Fibers towards Sustainable Power over Fiber Systems

Alicia Fresno-Hernández [†] , Marta Rodríguez-Guerra [†] , Roberto Rodríguez-Garrido [†] and Carmen Vázquez [†,*] 

Leganés Campus, School of Engineering, Electronic Technology Department, Carlos III University of Madrid Calle Butarque 15, 28911 Leganés, Madrid, Spain
* Correspondence: cvazquez@ing.uc3m.es
† These authors contributed equally to this work.

**Abstract:** This paper presents a transition taper for coupling light between optical fibers with different geometries and refractive index profiles used in Power over Fiber (PoF) systems. Global energy efficiency and costs are critical parameters when delivering high power to remote areas. High-power lasers have maximum coupling for large core fibers, while widespread multimode optical (OM1) fibers used in optical communications are cheaper. We study the optical losses between large core fibers (200 µm) and OM1 fibers (62.5 µm) theoretically and experimentally. We demonstrate that improvements of 2 dB can be obtained by adding the new tapered structure to the system, compared to the direct splice between both fibers. There is good agreement between measured and calculated loss values using a new Gaussian loss model to describe splices between tapered and straight fibers. The fabrication of the transition taper is also described. We also measure the numerical aperture (NA) changes in the downtaper zone and demonstrate that the lower the NA of the input light, the higher the efficiency improvement.

**Keywords:** optical fiber; energy efficiency; Power over Fiber; fiber coupling; tapers





## 1. Introduction

A Power over Fiber (PoF) system feeds a remote node with light transmitted through optical fibers. Generally, it consists of a high-power laser (HPL) as the emitter, an optical fiber as the transmission line, and a photovoltaic power converter (PPC), which converts the light into the electrical power that feeds the load. PoF is a good technique to send energy to distant points because the optical fiber has low weight and attenuation, is safe in places with high explosion risks, is immune to electromagnetic interference, and has good galvanic isolation. Therefore, this technology is useful in hazardous areas with high voltage and high electromagnetic activity [1], to feed medical devices [2] and sensors [3,4], or to monitor passive optical networks [5], among others. One key parameter in PoF systems is the overall energy efficiency, which is typically no more than 10% electrical to electrical for energy delivery to hundreds of meters on multimode (MM) fibers [6] or double-clad fibers [7]. This efficiency depends on the optical fiber losses and conversion efficiencies in the HPL and the PPC. Usually, there is a trade-off when selecting the HPL wavelength between increasing PPC efficiency, which is still under study [8], or reducing optical fiber attenuation, which depends on the feeding link length. However, the efficiency of optical coupling can also limit the PoF system's performance [1].

One way to improve optical coupling efficiency is through the use of tapers. Several theoretical studies on taper structures in single-mode (SM) fiber analyze how to achieve adiabatic tapers [9]. The theoretical study of tapers in multimode (MM) fiber based on mode propagation models is more complicated, and although there are some works [10–13], including those describing low losses in lantern designs [14], its complexity justifies further

study. A ray-tracing model has the potential to track the propagation of the guiding rays along the tapered fiber and the NA variations depending on the launching conditions, providing similar results to a mode propagation model under overfilled launch conditions (OFL). These models can also provide information on loss calculations when dissimilar fibers, such as those used in PoF systems, need to be connected. Multimode tapers have many applications, such as humidity sensing [15], and strain and temperature sensing [16]. However, in these cases, the taper is intended to expand the field to the cladding, increasing the evanescent field to make it more sensitive to external environmental changes. Therefore, the smaller the waist, the better the increase in sensitivity, but the worse the taper losses. This is the opposite target of improving PoF system energy efficiency by reducing overall optical losses through this structure.

In this work, we propose a transition downtaper to move from step-index (SI) 200 μm optical fiber to GI 62.5 μm optical fiber, so that the coupling is maximized, increasing efficiency when the cheaper, lower attenuation, and broadband optical OM1 fibers are used. The theoretical losses are estimated for different launching and propagating conditions, including a novel Gaussian loss model to describe splices between tapered and straight fibers. The transition downtapers are designed and manufactured, and later spliced to GI MM fibers. Their losses are characterized by different launching conditions using multimode and single-mode optical fibers, demonstrating the achievable loss improvements. The NA evolution through the downtaper section is also measured and analyzed.

## 2. Principles and Theoretical Study

Figure 1 schematically illustrates a PoF system requiring a connection between dissimilar fibers in the central office because there is a mismatch between the HPL pigtail and optical power delivery fiber. This is the case in the transition between an SI 200 μm optical fiber and GI OM1 62.5 μm optical fiber to reduce the cost and transmission losses of PoF systems. OM1 is a standard for multimode fibers defined by the Telecommunications Industry Association (TIA) and has been widely deployed in building and some outdoor applications for many years. There are many manufacturers providing this type of fiber, and its broad deployment in specific applications and the existence of many manufacturers make the cost of this fiber less expensive than others not standardized for a specific application, such as 200 μm step-index fibers. For example, the cost of 200 μm step-index fibers can be up to around 5 times more expensive, according to [17].

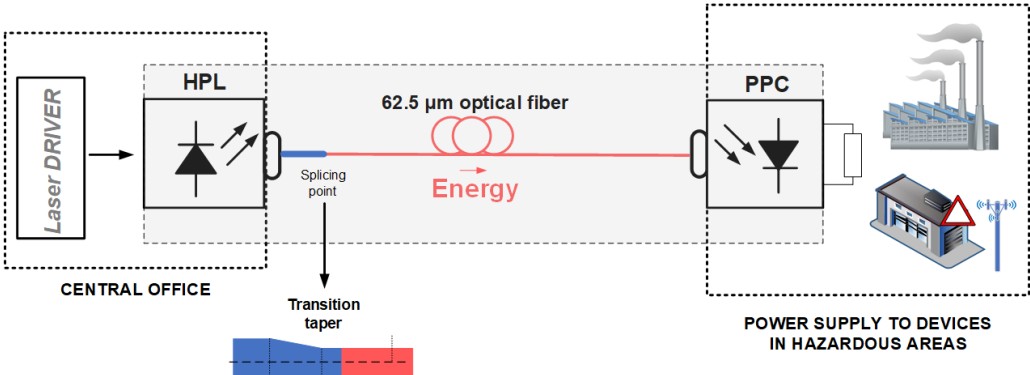

**Figure 1.** Schematic of the proposed structure and application.

In this section, we estimate the theoretical losses in joints of dissimilar fibers for different propagation conditions. We also define the notation and describe the structure of the proposed transition taper.

### 2.1. Multimode Fiber to Multimode Fiber Transition

The coupling between fibers of different characteristics always involves a certain amount of losses. In this work, the theoretical study of this phenomenon will be approached

in two different ways: Firstly, by considering an OFL, in which all existing modes in the multimode fiber will be equally excited by the source, and it will be assumed as an approximation that they suffer a uniform loss during propagation (uniform loss model). Secondly, considering a situation in which higher-order modes will eventually carry less power than the lower ones when arriving at the transition to the next fiber, following a Gaussian-shaped distribution (Gaussian loss model).

### 2.1.1. Uniform Loss Model

In the first approach, the power carried by the first fiber is uniformly distributed among all the modes. Within this OFL condition, the losses involved in the transition (expressed as a positive number) are related to the change in the number of guided modes in the following way [14]:

$$L = -10 \cdot \log \frac{N_2}{N_1} \tag{1}$$

where $N_1$ and $N_2$ are the mode numbers of the first and second fibers, respectively. If the number of modes involved is sufficiently large, the number of modes N is given by [18]:

$$N = a^2 k_0^2 n_1^2 \Delta \frac{x}{x+2} \tag{2}$$

where a is the core radius of the fiber, $k_0$ is its wave number in the free space, $n_0$ and $n_1$ are the refractive indices of the cladding and core (the maximum value in the case of GI fibers), and $\Delta = \frac{n_1^2 - n_0^2}{2n_1^2}$ is the refractive index contrast. The power index, x, defines the fiber refractive index profile, n(r), given by:

$$n(r) = n_1 \begin{cases} \sqrt{1 - 2\Delta \left(\frac{r}{a}\right)^x}, & \text{if } 0 \leq r \leq a \\ \sqrt{1 - 2\Delta}, & \text{if } r \geq a \end{cases} \tag{3}$$

where r is the radial coordinate; x is infinite for SI fibers and equals 2 for parabolic GI fibers.

When combining Equations (1) and (2), we arrive at the following expression:

$$L = -10 \cdot \log \frac{(a_2 NA_2)^2 (x_1 + 2) x_2}{(a_1 NA_1)^2 (x_2 + 2) x_1} \tag{4}$$

where $NA_i$ is the maximum numerical aperture of fiber i given by:

$$NA_i = \sqrt{n_{1i}^2 - n_{0i}^2} \tag{5}$$

The losses in Equation (4) are a combined effect of the difference in radius, numerical aperture (NA), and refractive index profiles of the fibers [19], expressed in dB, and are given by:

$$L = -10 \left[ \log \frac{(a_2)^2}{(a_1)^2} + \log \frac{(NA_2)^2}{(NA_1)^2} + \log \frac{(x_1 + 2) x_2}{(x_2 + 2) x_1} \right] \tag{6}$$

If the receiving fiber has a larger NA than the emitting one, i.e., $NA_2 > NA_1$, a non-physical gain is obtained. Thus, whenever this is the case, the corresponding term in the calculation of the losses must be considered to be 0, a criterion that will also be applied to the first term whenever $a_2 > a_1$. On the other hand, if the first fiber has a larger NA than the second one, $NA_1 > NA_2$, the formula predicts some reasonable losses in the transition. The NA in Equation (5) is defined in the context of ray optics as:

$$NA = n_p \cdot \sin\theta_{max} \tag{7}$$

where $\theta_{max}$ is the maximum incident angle of an external ray that can be guided when entering the fiber (within the conditions of total internal reflexion) and $n_p$ is the refractive

index of the previous medium from which the ray comes. In this sense, the NA defines the angular acceptance of the fiber. Nevertheless, in the case of straight fibers excited with sufficiently wide sources, the NA parameter is also a measure of the divergence of the beam coming out of it, provided that the medium surrounding both sides is the same.

In the case of Equation (6), it is implicitly assumed that the NA of the emitting fiber, $NA_1$, is a measure of the divergence of the light beam coming out of it, while the NA of the receiving one, $NA_2$, is a measure of its angular acceptance.

Along with the calculations involving numerical apertures throughout the rest of this paper, it must be kept in mind that, during a transition from fiber 1 to fiber 2, $NA_1$ and $NA_2$ will refer to the divergence of the output beam of the first fiber and the angular acceptance of the second, respectively.

### 2.1.2. Gaussian Loss Model

In a second approach, a Gaussian steady-state power distribution will be assumed to arrive at the transition from fiber 1. This Gaussian-shaped beam model [20] is based on a steady-state situation of the traveling light similar to that originated by a restricted launch condition with different power in each mode and considering that higher order modes carry less power than the lower ones. This model has been used to calculate losses in splices with good results [21]. A transmission function for each spatial point of the splice based on the NA of both fibers is introduced. This function, t(r), defined as the ratio of the power distribution being accepted by fiber 2, $p_2(r)$, and the one exiting fiber 1, $p_1(r)$, is given by [21]:

$$t(r) = \frac{p_2(r)}{p_1(r)} = \begin{cases} 1 + Qp_0 - p_0^Q, & \text{if } Q < 1 \\ 1, & \text{if } Q \geq 1 \end{cases} \tag{8}$$

where r is the radial coordinate of the point of the transition and $p_0$ is the fraction of the power peak arriving at the splice that has been chosen to define the NA. The quantity Q is given by [21]:

$$Q = \frac{\tan\left(\arcsin \frac{NA_2(r)}{n_2(r)}\right)}{\tan\left(\arcsin \frac{NA_1(r)}{n_1(r)}\right)} \tag{9}$$

where $n_1(r)$ and $n_2(r)$ are defined as in Equation (3) for both fibers involved in the transition, and $NA_1(r)$ and $NA_2(r)$ are the numerical aperture values at each point of the fiber cores:

$$NA_i(r) = n_{1i} \sqrt{2\Delta_i} \sqrt{1 - \left(\frac{r}{a_i}\right)^{x_i}} \tag{10}$$

This quantity is approximated by $Q = \left(\frac{NA_2(r)}{NA_1(r)}\right)^2$ in [20,21] but when the mismatch in the NA value is high, this approximation is not accurate, so in this work, it has not been used.

In addition to this, a weighting function to reduce the amplitude of the steady-state Gaussian distribution as a function of radius is required since only higher-order modes of propagation are significant near the core-cladding interface. Considering the generalized near-field power expression corresponding to the end face of a fiber of power index x in the uniform power distribution model [22] as this weighting function,

$$P_u(r) = P_0 \left[1 - \left(\frac{r}{a}\right)^x\right] \tag{11}$$

For the case of uniform power across the cone of radiation defined by the numerical aperture where $P_0$ is proportional to $\Delta$, $n_1$, and input power, the total power distribution for the Gaussian model at r can be calculated as in [20] and is given by:

$$P(r) = P_0 \left[1 - \left(\frac{r}{a}\right)^x\right]^2 \tag{12}$$

where $P_0$ is the amplitude of the field at $r = 0$.

The total transmission value, T, is obtained when multiplying $t(r)$ by the power arriving at point $r$ of the transition, integrating over the area of the core overlap and dividing by the total power arriving from fiber 1. It is given by:

$$T = \frac{P_2}{P_1} = \frac{\int_0^{2\pi} \int_0^{R_t} t(r) \left[1 - \left(\frac{r}{a_1}\right)^{x_1}\right]^2 r \, dr \, d\theta}{\int_0^{2\pi} \int_0^{a_1} \left[1 - \left(\frac{r}{a_1}\right)^{x_1}\right]^2 r \, dr \, d\theta} \tag{13}$$

where $R_t$ is the radius of the transition region. For a transition between axially symmetric fibers, this integral can be simplified to:

$$T = \frac{\int_0^{R_t} t(r) \left[1 - \left(\frac{r}{a_1}\right)^{x_1}\right]^2 r \, dr}{a_1^2 x_1^2 / 2 (x_1 + 1)(x_1 + 2)} \tag{14}$$

This expression can be used to calculate the intrinsic losses, which are given by:

$$L = -10 \cdot \log \frac{\int_0^{R_t} t(r) \left[1 - \left(\frac{r}{a_1}\right)^{x_1}\right]^2 r \, dr}{a_1^2 x_1^2 / 2 (x_1 + 1)(x_1 + 2)} \tag{15}$$

Additional extrinsic losses caused by the mechanical alignment of the fibers, such as the offset between the fibers, angular misalignments, etc., increase this theoretical value.

### 2.2. Fiber Tapers

Making a taper in fiber means reducing the diameter of its cladding ($W_{cl}$) and core ($W_{co}$) by means of stretching and heating the fiber (see Figure 2). For large $W_{co}$ fibers, there are many guided modes with different field profiles. In the downtaper transition zone ($Z_b$), the core diameter decreases, and some modes previously guided in the core now become guided through the cladding waist ($W_{w,cl}$), with the external medium acting as the new cladding [14,23]. If the fiber needs to be recoated, the coating and its refractive index must be chosen to ensure good guidance of the light transferred to the cladding along the downtaper zone and the waist, to prevent additional losses during its propagation.

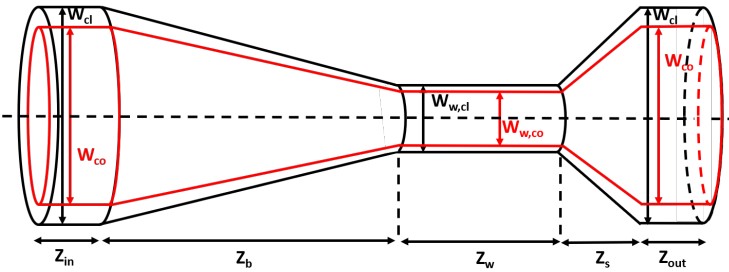

**Figure 2.** Schematic of a taper with $W_{cl}$ and $W_{co}$ representing the cladding and core diameters of the fiber, and $W_{w,cl}$ and $W_{w,co}$ representing the cladding and core diameters of the taper waist. $Z_b$ is the length of the downtaper transition zone, $Z_w$ is the length of the taper waist, and $Z_s$ is the length of the up-taper transition zone. $Z_{in}$ and $Z_{out}$ represent the lengths of the input and output zones of the fiber.

If the $Z_b$ transition is not gradual enough, the modes are coupled to evanescent higher-order modes that are radiated and generate losses. However, if the parameters are optimized and a sufficiently gradual transition is achieved, this evanescent field is reduced. At the up-taper transition region ($Z_s$), the modes are able to recouple to the core, achieving an adiabatic taper and drastically reducing losses.

An alternative way of explaining the propagation of light inside a tapered structure is by using ray tracing theory. Inside the taper, the surface where reflection and refraction occurs is tilted with respect to the fiber axis, and the traveling angle of the rays varies and grows with each reflection.

Considering the whole downtaper zone and defining the taper ratio $R = \frac{W_{co}}{W_{w,co}}$, the final output beam divergence is given by [24]:

$$NA_{output} = R \cdot NA_{input} \tag{16}$$

Because of the tilt of the taper surface, the maximum total internal reflection angle changes, altering the angular acceptance of the fiber. When using the NA parameter to describe this effect, the effective angular acceptance of the taper can now be obtained [25]:

$$NA_{t,acc} = \frac{1}{R}NA_0 \tag{17}$$

where $NA_0$ represents the original fiber NA as defined in Equation (5), $\theta_{max} = \arcsin\frac{NA_{t,acc}}{n_1}$ is the maximum incidence angle allowed on that point of the taper, and $n_1$ is the refractive index of the fiber core.

The angle of incidence of some of the rays will eventually exceed the decreasing $NA_{t,acc}$ of the fiber at some point, causing them to travel to the cladding of the fiber (i.e., those related to the higher-order modes). However, if the downtaper zone is sufficiently gradual, these rays will later re-enter the core in the upper zone, resulting in no losses (i.e., no coupling to evanescent modes).

### 2.3. Tapered Multimode Fiber to Multimode Fiber Transition

The difference in core dimensions between two dissimilar fibers is one of the main factors contributing to global losses in a transition [21]. To reduce these losses, we propose decreasing the core dimensions of the first fiber through an adiabatic taper, making it match the core of the next fiber ($a_1 = a_2 = \frac{W_{w,co}}{2}$). A waist with a constant radius will be left at the end of the downtaper zone, which will be cut at a certain point C and spliced to the next fiber, as shown in Figure 3.

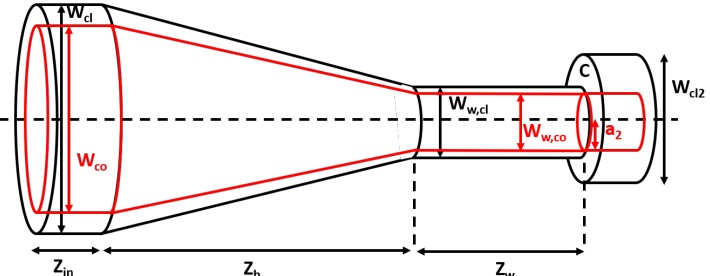

**Figure 3.** Scheme of the transition taper. From left to right: downtaper $W_{cl}$ fiber, downtaper transition zone of length $Z_b$, taper waist of length $Z_w$ cut at point C, and splice with a $W_{cl2}$ fiber.

According to Equation (16), the output numerical aperture of the first fiber will increase (maintaining its value along the waist until the splicing position). This increase in numerical aperture will predictably result in a loss increment in the transition due to the numerical aperture mismatch. However, it will be shown that the exchange of loss has a positive overall effect compared to a direct splice.

The equations derived in the previous sections can be used to analyze the losses in the transition from a tapered SI fiber to a straight parabolic index fiber, which is the case considered in this work. Expressions (4) and (15), from the uniform and Gaussian loss models, respectively, are combined with the expression for the output beam divergence ($NA_1 = NA_{output}$) from the taper given by Equation (16). Taking $x_1 = \infty$ and $x_2 = 2$, and

setting the same core dimensions for both fibers, $a_1 = a_2 = \frac{W_{w,co}}{2} = a$, the losses after the taper with both models are given by:

$$L_{t,Uniform} = -10 \cdot \log \frac{NA_2^2}{2(R \cdot NA_{input})^2} \tag{18}$$

$$L_{t,Gaussian} = -10 \cdot \log \frac{\int_0^a \left(1 + Qp_0 - p_0^Q\right) r \, dr}{a^2/2} \tag{19}$$

where the parameter Q is now given by:

$$Q = \frac{\tan\left(\arcsin \frac{NA_2(r)}{n_2(r)}\right)}{\tan\left(\arcsin \frac{R \cdot NA_{input}}{n_1}\right)} \tag{20}$$

### 3. Design and Experimental Results

#### 3.1. Design and Manufacturing

We designed a transition taper from $W_{co}/W_{cl}$ fiber to $W_{w,co}/W_{w,cl}$ fiber. We fixed the values $Z_b, Z_w, Z_s$ (see Figure 2) to obtain an adiabatic taper. Following the conclusions of [23], and after a significant optimization process of both taper parameters and heat parameters of the FSM-100P+, to extend their results to the MM case by manufacturing tapers of different dimensions, we propose a linear adiabatic taper with $Z_b$ = 9 mm, $Z_w$ = 7 mm, and $Z_s$ = 1 mm. The tapers were made using a Fiberguide AFS200/220/320Y multimode fiber, 200/220 µm core/cladding diameter. As the fiber's core/cladding ratio is constant when a full taper is conducted [26], we manufactured tapers with $W_{w,cl}$ = 68.75 µm to obtain $W_{w,co}$ = 62.5 µm, which allowed us to splice it with the OM1 fiber. The process of stretching and heating the fiber to manufacture the transition taper was done with a Fujikura FSM-100P+ fusion splicer. The manufactured taper shown in Figure 4 only has 0.4 dB loss.

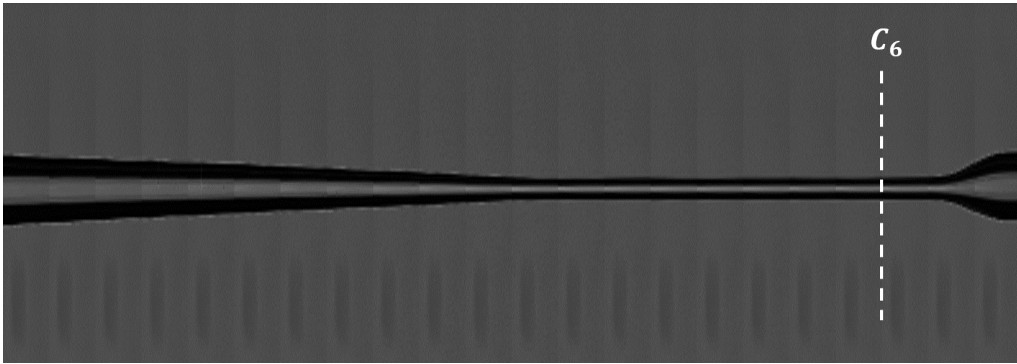

**Figure 4.** Experimental image of an adiabatic taper; $C_6$: cutting point.

After cutting the adiabatic taper in the waist in $C_6$, and before splicing, we checked the core width. We confronted an OM1 fiber, whose cladding (A) was 125 µm, with our taper, using the lateral view imaging of the splicer, see Figure 5a. By processing this image with Matlab, we determine how many pixels correspond to A and B; thus, we can obtain the size of B in µm, which is the cladding of the waist of our taper. In order to measure the core diameter, we remove the OM1 fiber and use the frontal view image option of the splicer, see Figure 5b. Following the same procedure, we know C = B and we obtain D, knowing C. The results obtained for a taper with $W_{w,cl}$ = C = 68.75 µm are D = 62.5 $\pm$ 3 µm. For the same taper, the process was repeated 20 times to ensure a realistic error estimation. The NA change is proportional to R. In our case, from R = 3.2 and $NA_{input}$ = 0.22, we obtain $NA_{output}$ = 0.7 using Equation (16).

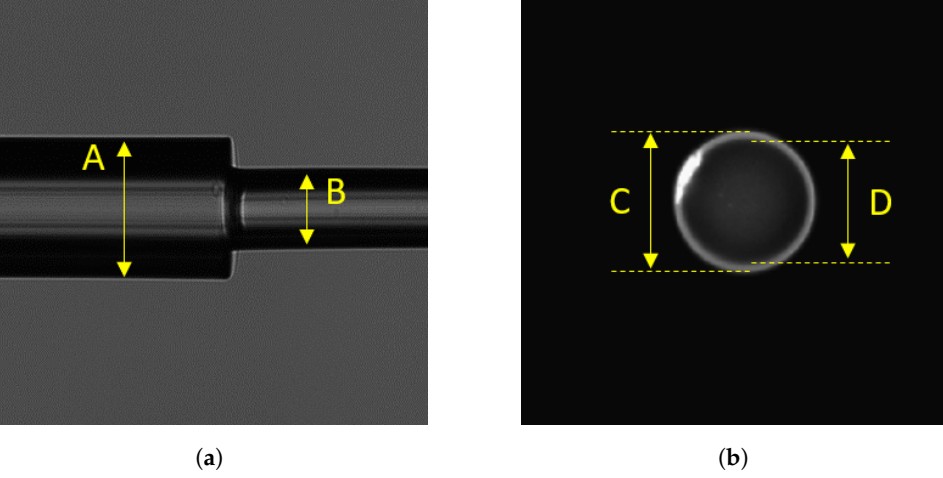

(**a**)                    (**b**)

**Figure 5.** (**a**) Lateral view imaging of the splicer. Left fiber: OM1 (62.5/125 µm), right fiber: taper. A: 125 µm B: $W_{w,cl}$. (**b**) Frontal view of the splicer. C: $W_{w,cl}$, D: $W_{w,co}$.

### 3.2. Measurements of the Output NA Evolution in the Downtaper Transition Zone

As we described in Section 2.2, the output numerical aperture increases after light passes through the downtaper zone. The evolution of NA was experimentally determined by making different cuts in the taper structure with a cutting machine (Fujikura CT-101) and analyzing the light coming out of the fiber.

Different tension values were chosen for each cutting zone shown in Figure 6.

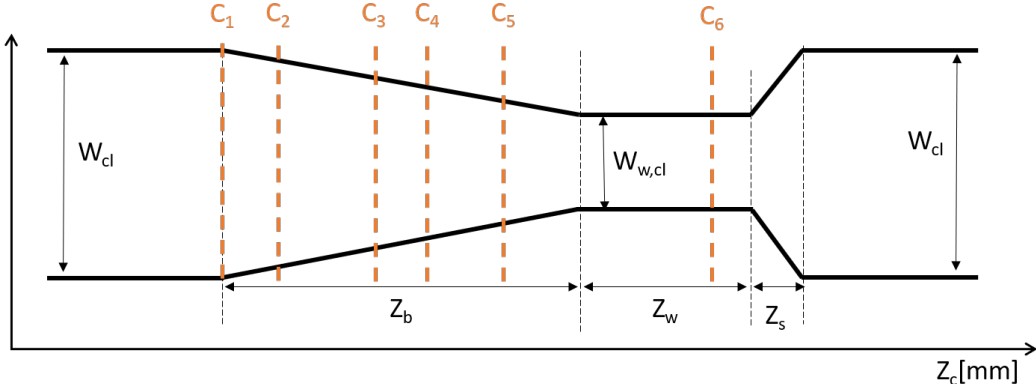

**Figure 6.** Scheme of cuts in the downtaper transition zone and waist.

The length and cladding diameter of the different cuts are shown in Table 1. For each cut shown in Figure 6, a new taper was made. The Fujikura FSM-100P+ fusion splicer has an option that is capable of measuring the taper by moving the splicer motors; we used this option to measure the distance in the downstream area we were interested in, and made a mark with a permanent marker. Afterwards, we made a cut with the CT-101 exactly on the mark and checked the position by taking images with the end view of the fusion splicer, and applying the A, B, C, and D procedures explain before. So the method used to measure the cladding diameter was imaging, as explained in Section 3.1, whereas the length dimension, $Z_c$ (see Figure 6) was estimated afterwards by trigonometry.

**Table 1.** NA measurements.

| #Cut | $Z_c$ Length (mm) | Cladding Diameter (µm) | Taper Ratio | Theoretical NA * | Beam Profiler Measured NA | Integrating Sphere Measured NA |
|------|------|------|------|------|------|------|
| C1 | 0 | 220 | 1.000 | 0.22 * | 0.22 ± 0.04 | 0.24 ± 0.02 |
| C2 | 1.44 | 196 | 1.124 | 0.25 | 0.25 ± 0.04 | 0.27 ± 0.02 |
| C3 | 3.75 | 157 | 1.400 | 0.31 | 0.28 ± 0.04 | 0.31 ± 0.02 |
| C4 | 4.54 | 144 | 1.530 | 0.34 | 0.32 ± 0.04 | 0.35 ± 0.02 |
| C5 | 6.43 | 112 | 1.964 | 0.43 | 0.41 ± 0.04 | 0.43 ± 0.02 |
| C6 | 14 | 69 | 3.200 | 0.70 | 0.55 ± 0.04 | 0.66 ± 0.02 |

* From the AFS fiber datasheet. The rest of the column was derived from this one using Equation (21).

In order to measure the NA for each cut, two different methods are used [27]: a planar detection array (in our case, a beam profiler, i.e., BeamOn WSR from Duma Optronics), and an angular scanning (in our case, an integrating sphere from Thorlabs). When using the beam profiler, it was placed at two different positions from the fiber tip, separated a distance of Z as shown in Figure 7.

The output beam diverged so the beam width was different in each position [28]. By taking the beam widths of the different profiles, $W_1$ and $W_2$, the output angle is given by:

$$\theta = \tan^{-1}\left(\frac{W_2 - W_1}{2 \cdot Z}\right) \tag{21}$$

In Figure 8, the facet of the cut taper and the output beam profiles at two different distances for the different cuts are shown. The bright circle is the facet of the taper, and the dark circle behind is just the air of the temporary connector in which the taper must be placed to be observed. The vertical profiles from these beams ($W_1$ and $W_2$) are used to calculate the angle using Equation (21).

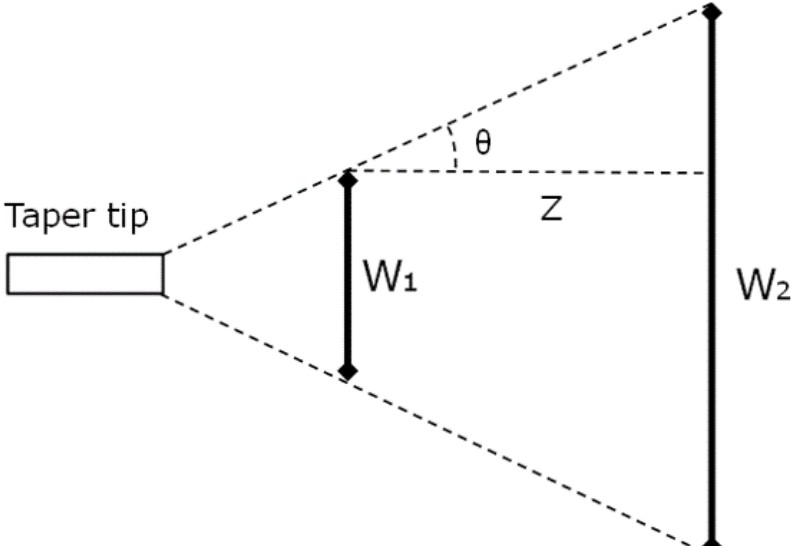

**Figure 7.** NA measurement set-up. The fiber tip was placed at two distances from the beam profiler.

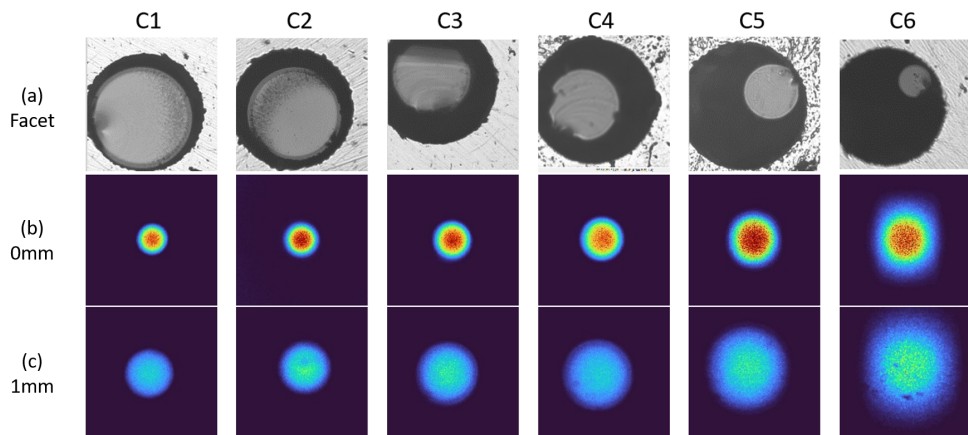

**Figure 8.** Real images of each cut $C_1$–$C_6$: (**a**) facets of the cut tapers obtained with a fiber inspector. (**b**,**c**) beam profiler images of the field with the tip of the fiber (**b**) 0 mm (**c**) 1 mm.

When using the integrating sphere, the cut taper is placed into a temporary connector that is fixed in a rotatory stage, as shown in Figure 9.

Then, we place the integrating sphere at 45 mm from the fiber tip to be able to assume far-field measurements. By rotating the taper tip, we obtain the power data as functions of the rotation angle, as shown in Figure 10, for different cuts.

Using the 5% threshold [29], the output NA values are determined for all cuts. Table 1 shows the measurements obtained from both methods. The results show that the output NA increases along the downtaper region, with the results fitting very well with the theoretical expectations. Although both experimental results (the ones made using the beam profiler and the integrating sphere) fit very well with the theoretical expectations, they obviously have an associated error. In the case of the beam profiler, as we stated in the paper, to make the measurement, we had to take images of the profiles at two different distances, which usually, for us, were 0 and 1 mm. These distances must be fixed by hand; this can be difficult and result in some inaccuracies, so we considered a possible human error of 0.1 mm in the distance, which corresponds to an error of 0.04 in the NA. So, as it can be seen, this is pretty critical. Moreover, when we measure with the beam profiler, we make the measurements at a distance below the Rayleigh limit [30], meaning we are not in the far field and the measured angle will always be lower than the angle measured in the far field. This can also pose a problem in our case. On the other hand, with the integrating sphere, we must also measure the angles of rotation by hand, but it is easier to achieve better resolution, and the error in the NA is only about ±0.02, which is better than in the previous case. Furthermore, as we place the fiber at a distance of 45 mm, higher than the z of Rayleigh in our case (36.5 mm), it allows us to measure in the far-field region, avoiding any problems due to that. For these reasons, although the beam profiler measurements were useful in providing clear images of the NA increase in the downtaper zone, we decided to use the integrating sphere to make the measurements, as a consequence of the reasons explained above.

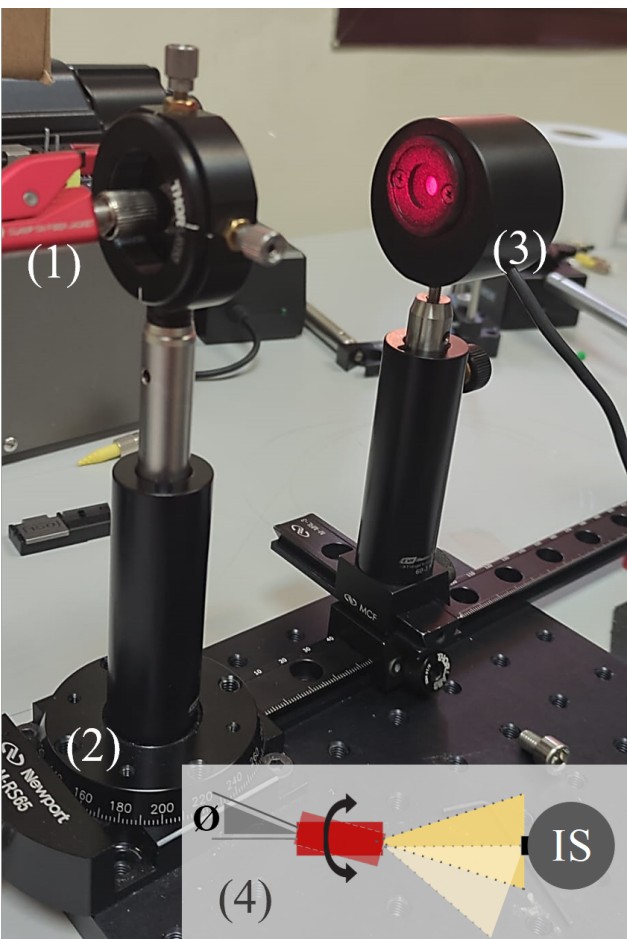

**Figure 9.** Experimental setup for measuring the NA with the integrating sphere, with a red laser used as light source as an example. (1) Temporary connector in which the taper tip is placed. (2) Rotatory foot that allows moving the temporary connector and measuring the angular displacement. (3) Integrating sphere (IS). (4) Top view of the system.

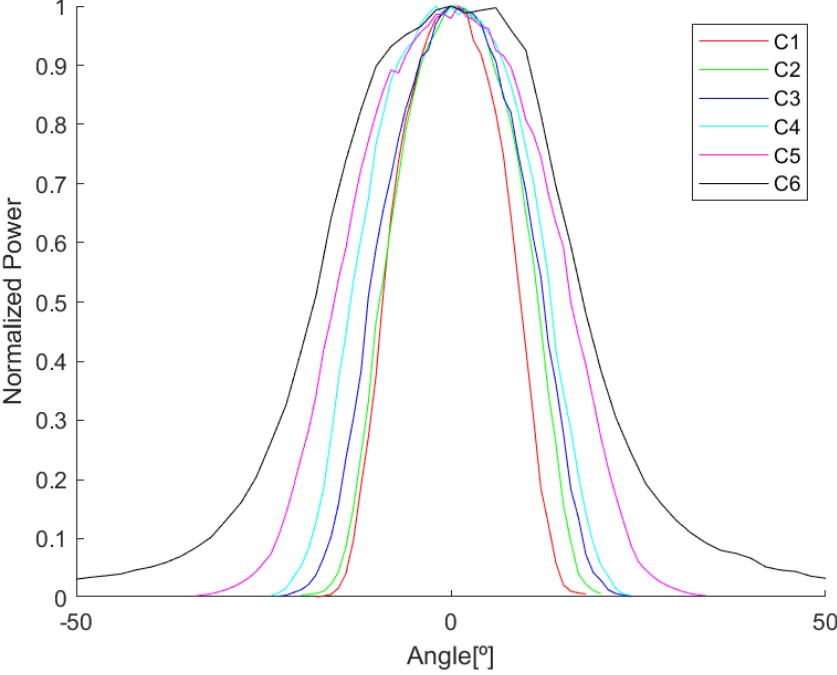

**Figure 10.** Normalized power profile measurements versus angles for each cut taper $C_1$–$C_6$.

### 3.3. Measurements of Loss Improvement Using the Transition Taper with MM Injection

The measurement set-up shown in Figure 11a was used for characterizing the manufactured transition tapers. The input source is an HPL with a central wavelength of 860 nm and optimized coupling to 200 μm core fibers with NA = 0.22. For security reasons, we introduced a variable optical attenuator (VOA) from UCC, which gave us a lower input power in the system without compromising the stability of the laser. We also added a switch (SW) to prevent the laser light from passing through the system while we worked on the splicer. Working with low power levels allowed us to use PDA100A2 Thorlabs photodetectors (PDs) capable of measuring in the range from 320 to 1100 nm. To have a self-referenced setup that accounts for laser power variations, we also introduced a 50:50 coupler. One of the branches of the coupler went to PD 2, and the other went to PD 1 after passing through our fiber under test (FUT). The photodetectors were identical, and both signals were measured simultaneously using a two-channel data acquisition card (DAQ). All of the aforementioned components involving optical fiber (VOA, SW, coupler) were manufactured with Thorlabs FG200UCC fiber, since it is cheaper and meets the laser-coupling conditions. However, at the time of manufacturing the tapers, this fiber had double-cladding, which hindered the cutting process with CT-101. For this reason, it was finally decided to make a FUT, as shown in Figure 11b. At the beginning of the FUT, we utilized FG200UCC fiber to minimize losses when coupling it with the rest of the components, including the coupler. Next, we fused this fiber with the AFS200/220/320Y to work more comfortably with it, as the cutting process for this fiber is easier. Therefore, our tapers were manufactured in AFS fiber rather than in UCC, as the rest of the system. Two different FUTs are characterized (see Figure 11b,c). The losses obtained for the transition from UCC fiber to AFS fiber are around ∼0.2 dB. Taking this into account, we can obtain the loss improvement when using a transition taper by subtracting the power acquired after characterizing FUT 1 in Figure 11b and the power acquired after FUT 2 in Figure 11c. The summary of the results can be found in Table 2.

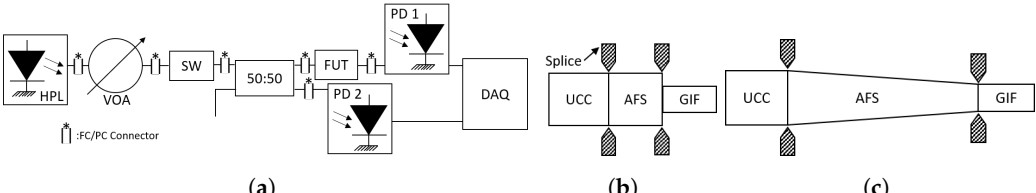

(**a**)          (**b**)          (**c**)

**Figure 11.** (**a**) MM measurement set-up: 860 nm HPL and MM injection. (**b**) Fiber under test 1 (FUT $1_{UCC}$): AFS-GIF direct splice. (**c**) FUT $2_{UCC}$: AFS-GIF splice with transition taper in between. VOA: Variable optical attenuator, SW: Switch, 50:50 coupler, PD: photodetector, DAQ: Data acquisition card. UCC and AFS 200 μm core fibers and GIF OM1 fiber.

**Table 2.** Summary of the results of losses for MM launching.

| Type of Injection | Experimental (Integrating Sphere) | Theoretical (Gaussian–Uniform) |
|---|---|---|
| Input Taper NA | 0.24 | 0.22 |
| Output Taper NA | 0.66 | 0.70 |
| AFS-GIF Direct Splice (dB) | 10.88 | 11.18 ** –13.11 ** |
| AFSTaper-GIF Splice (dB) | 8.91 | 8.77 ** –11.13 ** |

** Calculated using the NA from the commercial fiber datasheets and the previous theoretical formulas, with a threshold of $p_0 = 0.1$.

If we compare the case of the direct splice of dissimilar MM fibers with the case of the transition taper in between, there is a total improvement of 1.97 dB in the transmission. These results are further discussed in Section 4.

Regarding the choice of the threshold $p_0$ for the theoretical calculations, some comments need to be made. Initially, we followed the directions of [29] and measured the

output NA values summarized in Table 1 with a threshold of 5%. However, for the theoretical estimations, we used the theoretical output NA values obtained from the fibers data sheets, which did not specify a threshold. As $1/e^2 \approx 13.5\%$ is also frequently used as the threshold value for these purposes, we decided to assume $p_0 \approx 10\%$ for the theoretical estimations. This value gave good results that matched well with the experimental losses obtained and was also one of the suggested values in the reference paper for the Gaussian model [20].

Nevertheless, some additional calculations were made for different $p_0$ values that revealed that in the taper case, this parameter is significant, causing the loss estimation of the Gaussian loss model to change by approximately 1.5 dB in the 5–13.5% threshold range. This sensitivity is expected since in the taper case, more rays arrive at the transition point with a critical angle, and losses depend much more on the NA range than in the direct splice. Thus, the parameter $p_0$ must be handled with care in the Gaussian model, as results in some scenarios can be substantially affected by its choice.

### *3.4. Loss Improvement Measurements with SM Fiber Injection*

The most limiting parameter at the time of reducing the losses in the transition appears to be the NA mismatch. To verify this, we carried out another experiment introducing a G.652 standard SM fiber at the beginning of the system (see Figure 12), in order to change the launch condition by starting with a lower input NA and, therefore, obtaining a lower output NA at the end of the transition taper. Using the integrating sphere we measured the NA before and after creating the taper in the AFS fiber and we obtained NA values of 0.17 and 0.44, respectively.

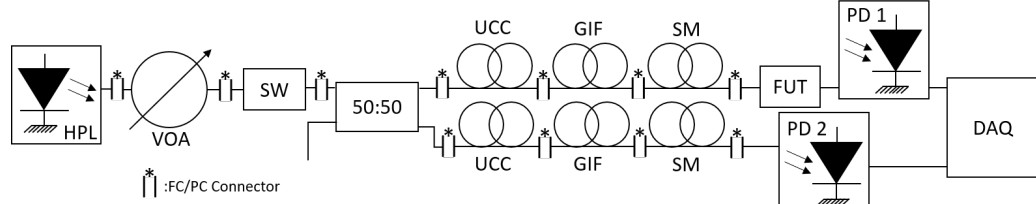

**Figure 12.** MM measurement set-up: 860 nm HPL and SM injection.

As explained in Section 3.3, all the components of the experimental system are made of UCC fiber. To introduce the SM fiber without making an abrupt transition (and avoiding back-reflections), we added three sections to both branches of the coupler: one of UCC fiber (200 μm), another of GIF fiber (62.5 μm), and one of the SM fiber (9 μm). Apart from this change, we used the same FUTs as in Figure 11b,c. The losses obtained in the case of the direct splice are 9.52 dB, and when we added the taper, they decreased to 6.58 dB. There is an improvement of 2.94 dB in this case, which makes it clear that decreasing the input NA helps improve the enhancement provided by the transition taper.

## 4. Discussion

To pass from a 200 μm core fiber to an OM1 fiber by means of a transition taper, we analyzed the losses that are produced when splicing two dissimilar fibers in different situations. In the direct splice, the theoretical models described in Section 2 predicted 11.18–13.11 dB of power losses by applying Equations (4) and (15), respectively. As can be seen in Table 2, the measured losses are around 11 dB, so the uniform excitation model overestimates the splice losses, whereas the Gaussian loss model prediction is more accurate. In the case of the splice with the transition taper, the theoretical models give an estimation of 8.77–11.13 dB from Equations (18) and (19), once again showing that the Gaussian loss model is more accurate when facing the experimental results obtained.

The expressions in Equations (8) and (12) seem to properly model the lower excitations of the higher-order modes, giving a more accurate perspective of what actually happens to

light in the transition between dissimilar fibers than the uniform model, helping to estimate the expected losses in splices between transition tapers and straight fibers for the first time.

The improvement in losses through the transition taper, mostly attributed to the disappearance of the radius mismatch, has been theoretically and experimentally verified. The input NA appears to be a critical parameter for the final losses of this type of system because the mismatch in NA grows after the taper structure while decreasing the first fiber radius to match the next one. With the MM injection and the taper structure, we obtain an experimental value of $NA_{output}$ = 0.66 (see Table 2), in which a decrease of 1.97 dB in losses is achieved with respect to the direct splice without taper. To prove that the losses reduction is higher if the input NA is lower, we changed the launching fiber to an SM one and demonstrated that this results in a smaller experimental $NA_{output}$ = 0.44 after the AFS taper, which enhances the losses reduction in the transition to the GIF to 2.94 dB. Although this SM fiber is outside of the scope of the article, as it would limit the power we can supply and, therefore, could never be proposed as a final system, it has allowed us to demonstrate how important the numerical aperture is in this system and for further applications with other types of fibers. The $NA_{input}$ value could be reduced even more, with the consequent increase in the performance of the transition taper structure in the coupling and the increase in the efficiency of the overall PoF system, where it could be used. Moreover, other types of tapers that are not linear, such as exponential and sinusoidal, could further improve our results, providing losses below 0.4 dB that will be analyzed in future works.

The losses of a PoF system based on an 860 nm HPL, with the highest PPC efficiency reported to date [8], and 800 m of UCC fiber, are 11.76 dB. With the system that we propose, which includes the HPL, a piece of UCC to improve the coupling of the laser, the transition taper structure (considering our best results in terms of losses), and 800 m of OM1 fiber, the losses are 11.3 dB, and the price, considering only the fiber cost [17], is around 80% cheaper. It is also important to point out that currently, there is OM1 dark fiber available for straightforward integration of the PoF infrastructure, while new digging is required for singular fibers such as UCC fiber. In conclusion, our proposed transition taper can be very useful when talking about distances of hundreds of meters, which are commonly used in hybrid systems with PoF on mobile network fronthauls [6].

## 5. Conclusions

PoF is a good technique to send energy and data to remote zones, especially in places with high electromagnetic activity or the risk of explosion, among other factors, and a key performance indicator is the overall efficiency of the system. HPL is typically coupled to large core multimode fibers with higher losses and costs than standard OM1 fibers. This work describes the design, manufacturing, and characterization of transition tapers as effective methods to reduce optical losses in dissimilar fiber transitions. The usefulness of transition tapers in systems with a transition from a large core fiber to another standard narrower fiber is shown, and a detailed study of the evolving NA and propagating modes was used to explain the remaining losses in these transitions. The higher the input NA, the lower the improvement the transition taper provides in the coupling. To prove this, we analyzed different set-ups and measurements with various launching conditions, achieving loss improvements of 2 dB in general MM launching conditions and up to 3 dB in restricted SM launching conditions. We proposed a Gaussian loss model to describe the splices between tapered and straight fibers; it properly estimates the experimental losses. These transition tapers can be useful in hybrid systems with PoF on mobile network fronthauls spanning hundreds of meters, using cost-effective OM1 dark fibers.

**Author Contributions:** Conceptualization, C.V. and A.F.-H.; data curation, R.R.-G.; methodology, R.R.-G. and A.F.-H.; software, M.R.-G. and A.F.-H.; formal analysis, A.F.-H. and M.R.-G.; investigation, R.R.-G., A.F.-H. and M.R.-G.; writing—original draft preparation, A.F.-H., M.R.-G. and R.R.-G.; supervision, writing—review and editing, resources, project administration, validation, and funding acquisition, C.V. All authors have read and agreed to the published version of the manuscript.

**Funding:** The research was supported in part by Agencia Estatal de Investigacion (PID2021-122505OB-C32), Research and Innovation Programme from Community of Madrid SINFOTON2-CM (S2018/NMT-4326), TEFLON (Y2018/EMT-4892), Ministerio de Asuntos Economicos y Transformacion Digital European Union–NextGenerationEU (TSI-063000-2021-135) and by FSE/FEDER funds and the FPU grant (FPU19/04133).

**Institutional Review Board Statement:** Not applicable.

**Informed Consent Statement:** Not applicable.

**Data Availability Statement:** The data underlying the results presented in this paper are not publicly available at this time but may be obtained from the authors upon reasonable request.

**Acknowledgments:** We acknowledge helpful discussions with R. Altuna-Pérez in the experimental set-ups.

**Conflicts of Interest:** The authors declare no conflict of interest.

## Abbreviations

The following abbreviations are used in this manuscript:

| | |
|---|---|
| NA | numerical aperture |
| PoF | Power over Fiber |
| HPL | high-power laser |
| PPC | photovoltaic power converter |
| MM | multimode |
| SM | single mode |
| GI | graded-index |
| SI | step-index |
| OFL | overfilled launch condition |
| FUT | fiber under test |
| AFS | AFS200/220/320Y MM fiber |
| UCC | FG200UCC MM fiber |
| VOA | variable attenuator |
| SW | Switch |
| PD | photodetectors |
| DAQ | data acquisition card |

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
