# Peer review of "Downtaper on Multimode Fibers towards Sustainable Power over Fiber Systems"

_photonics, doi:10.3390/photonics10050513_

Round 1

Reviewer 1 Report

The manuscript describes theoretically and experimentally a down-tapering of a step-index 200 µm core diameter multimode fiber in order to increase the coupling efficiency to a more standard graded-index OM1 multimode fiber. The theoretical part is quite well presented, and some questions arises from it but the experimental part needs more deep correction. Effectively, many things remain unclear even after reading a few times the text. Some other parts are also a bit suspicious as the way how they have been able to cut a portion of fiber that is less than 1 mm with a cleaver as a Fujikura CT-101. So, a good improvement on this part is needed. I hereafter, write down my different comment taken along the read of the manuscript :

Introduction :

The end of first paragraph and beginning of second don't link well. At first, we read information about coupling efficiency and then we start with tapers. To better link the two paragraphs, authors should mention that tapers may be a solution to improve the coupling.

Principles and theoretical study :

Authors mention that the use of a GI 62.5 µm optical fiber is more efficient and cheaper than using a SI 200 µm fiber. Ok. But there is no information supporting that statement. If the HPL laser providers mount a 200 µm fiber on their lasers is because there is a need for such fiber. Also, the GI 62.5 µm fiber is cheaper and more efficient, but what are the real differences in price and efficiencies ? Regarding price, are we 2 times, 10, 100 times cheaper or maybe just 1% ? Regarding efficiency, what is the difference ? Do we have less losses, and if yes, what are the numbers ? Please bring more detailed information as it is the main basis of the decision to use those OM1 fibers, but we have no information at explain it, why it would be better to use such OM1 fibers.

Line 67 : Overfilled Launch condition OFL has already been mentioned in line 38. Please, correct.

Lines 68-69 : Author mention that when all modes are equally excited (OFL) they all experience a uniform loss. However, each group of modes will have their own propagation constants and their own propagating losses. That's true the loss difference between them won't be huge, but there are differences. So, please correct that statement.

Equation 1 : If N2 is greater than N1, then L is <0, which has no sense as a negative loss could be interpreted as a gain. This is why in [14], they specify the equation "(expressed as a positive number)" to overcome this issue. So, please, correct.

Line 78 : Authors say that a is the radius of the fiber. To be correct, they should write that a is the core radius of the fiber. Please, correct.

Line 79 : (n1^2-n2^2) must be divided by 2x the square of core refractive index and not the cladding refractive index. Please correct.

Equation 3 : I have an issue with that equation and its explanation. Before that, it is mentioned that n1 is the core refractive index (line 78-79) whilst it is the maximum value of n1 within the core. Then, the refractive index profile should be dependent on that maximum n1 and not n(0) as there can be some misinterpretation between n1 and n(0) which are in fact the same ! Please, correct.

Equation 4 : I don't get why it is (xn + 2). It should be (xn + 1) as equation 2 shows. Please, explain or correct.

Equation 10 : n1 (maximum value) is missing to multiply the refractive index change part of the equation.

Equation 11 : In [19], the term in brackets is not squared. Why should it be here ?

Lines 142-143 : Authors write that the external medium is acting as the new cladding. That is true if the tapered fiber remains uncoated. But I guess that for a final product the tapered fiber will be recoated back to protect it. Does it change something on the theoretical model or the experimental results ?

Figure 2 : This figure generates a lot of concerns. First of all, it is not clear at all what is what. I understand that Zb zone is the down-taper zone but it is less clear what is exactly the Zw and Zs zones. Also, I understand that the blue lines are the core/cladding interface, but are the black ones the cladding diameter ? Also, from an optical point of view, I can't believe that such optical guide can guide light with total internal refraction using such important angles. So, please, explain in the text why you use such extreme angles as to better allow the reader to understand what happens. Improve the general description of your picture to understand any information quickly and accurately. Figure 3 is probably better explaining what you are dealing with.

Lines 162-163 : Authors write that the taper won't suffer losses due to the high-order modes leakage to the cladding re-entering the core in the up-taper zone. But this situation can remain true if the fiber is made of high purity fused silica as it shows a very low mode coupling coefficient. That's good as the present work deals with fused silica fibers. However, an issue could be if the core diameter approaches the cladding diameter, and thus the cladding-air (or coating) interface could be reached by this leaking light. So, please, give a word, an explanation of what is the situation for the configuration you propose with this 200 µm core fiber. Is any loss to be expected or not because the cladding diameter is too small to comfortably keep this leaking light in the cladding ? Give some inputs.

Design and experimental results :

Lines 208-210 : I have a concern about the cleaving process in order to measure the NA change within the down-taper transition. First of all, the distances are really short, and some of them are even under the minimum cut length of the CT-101, which is of 3 mm. For example, the cut from C4 to C3 is only 0.79 mm which is absolutely not feasible with a "tension and cut" machine as the CT-101. So, the only way to approach experimentally that measurement would be to take, reasonably, as much tapers as the cuts presented in figure 6 and to cut all of them at a different length. So, I need a clear and detailed procedure how these cuts were performed, otherwise it cannot be interpreted as a valid measurement.

Lines 224-226 : Authors say that they take into account that the taper facet is placed in front of a 1.7 mm protection window. But, what does it means "take into account" ? Is there any angle change due to some refraction ? What happens and, if some correction is needed, what it is ?

Figure 8 : We can see from the a) line that the taper end is inside a circular hole. However, I can't see any description of what this hole is. Please, describe. Also, this hole shows that there is a material that can reflect towards the CCD sensor the light that has been reflected by the CCD's protective window. What is the influence of these reflections on your measurement ?

Lines 227-228 : Authors write "the cut taper is placed into a temporary connector that is fixed in a rotatory stage". As it is an important setup used to measure the NA with an alternative solution, please provide a picture or scheme showing your measurement principle and setup used.

Line 231 : Authors claim they have followed the 5% threshold recommendation from [28]. However, in fiber optics, we usually follow the effective numerical aperture which refers to the diameter where the beam reaches 1/e^2 level. That is a 13.5 % threshold and not 5%. So, how do you link your measurement results (5 %) and the simulation NA you've provided. Is the 5% measurement more comparable to your simulation or is the 1/e^2 level more appropriate ? Please, comment.

Lines 234-236 : Authors decided to finally use the integrating sphere for the rest of measurements. However, they don't mention the reason to make this choice and discard the beam profiler. What are the reasons that made you decide to use only the integrating sphere ?

Figure 10 : We can see the setup used to measure the losses when injecting light to a configuration with and without tapering the AFS fiber. However, I don't get the information of why do you use a UCC fiber before the AFS. Is the fiber provided with the laser itself ? Please add the information in the manuscript.

Also, once the UCC is an attenuator (line 245) and another it is the fiber. Please, clarify.

Figure 11 : I have many problems with that figure and its description. We see that the SM (SM fiber) is spliced after the UCC+GIF fiber in both arms of the 50:50 coupler and is spliced at its output to an "FUT" fiber on upper arm. So, is FUT corresponding to AFS fiber ? If I look at this upper arm, I can understand that the SM fiber is spliced to an AFS fiber that is down-tapered before entering the photodetector. But this has no sense.. Why doing that after the SM and not before. I think that I have misunderstood what you did here and this is probably because things are not clear enough for the reader to quickly get what you did. Or, I have understood very well, but, in this case, the FUT (AFS ?) must be spliced and down-tapered before the SM fiber ? Only in discussion section, we learn that the SM fiber is after the AFS taper ! But, it's not clear at all on this figure. Make it much clearer as we don't get what you did here.

Also, as your manuscript deals with the transport of light energy necessary to power some remote electronics, why do you want to use SMF fiber that will limit the power ? Is it just because you want to show the tapering effect on smaller core diameter fibers splices ? If, yes, please let know the reader that you are out of scope regarding the power transmission but more on showing the ability of the solution you propose for another type of fibers and applications.

Discussion :

Lines 305 - 318 : Authors start a discussion about the cost of the proposed solution and its comparison to what has been published in [8]. Nevertheless, it is not clear where these 2'200 € haven been taken and what includes the price of 544 € they state. Are the man-effort and side-costs taken into account for this cost comparison ? If not, this cost comparison has no value as only the "components cost" is not very useful for such small difference. Take into account that setting up such solution from A to Z on the field will cost much much more than the prices that we can read.

Another concern is about the "exact" prices given as 544 €. This price won't be true all along the manuscript lifetime and, even, Euro currency may potentially disappear one day. It would be better to talk about halving the price or also a 30% of costs instead of an exact price.

Reviewer 2 Report

In paper, entitled, “Down-Taper on Multimode Fibers towards sustainable Power over Fiber Systems” the Alicia et al, presents a transition taper for coupling light between optical fibers with different geometries and refractive index profiles used in power over fiber systems. The results are interesting and well-summarized. I would recommend the manuscript to be accepted after minor revision. The following should be noted before acceptance:

1.      What are the specifications of HFL system used in Fig. 1?

2.      At sentence no. 193, the author claimed that tapered fiber shows a losses of 0.4 dB losses. I would like to ask what are optimum losses at which we can say that fiber is best tapered.

3.      In Fig. 5 (a), fiber A, a core seems to be damaged; does it affect the light propagation or enhance losses?

4.      The Figures resolution needs to be improved.

Reviewer 3 Report

Review of paper:

Title: Down-Taper on Multimode Fibers towards sustainable Power

over Fiber Systems

Authors: Alicia Fresno-Hernández, Marta Rodríguez-Guerra, Roberto Rodríguez-Garrido and Carmen Vázquez.

The paper is very difficult to read. There is lot of mathematical formulations which are not so important for understanding of principle of optical fiber taper. The precondition that  all modes are powered by the same energy is physical incorrect.

The results of theory and measured data are in perfect conformity for both precondition of uniform and Gaussian powered modes but without confirm with different OF taper parameters.

In our previous paper titled “Mathematical Model for Laboratory System of Bioluminescent Whole-Cell Biosensors with Optical Elements” published in Journal of Biosensors and Bioelectronics 2018, Vol.9. Issue 1, pp. DOI: 10.4172/2155-6210.1000250 is present numerical calculation of transmission by OF taper. This value depends on shape of taper wall and lengths.

In case of present paper, the calculations are made for linear changes of diameter of OFT with lengths what is practically not possible to make. In paper is not analysed results as a function of length of taper.

For these reasons I cannot recommend the article for publication in present form.

Round 2

Reviewer 1 Report

Authors have satisfactorily treated all my comments and the manuscript is now more readable and its quality has increased. I can confirm that I recomment it for publication.

Author Response

Thank you for your comments and your final opinion on the manuscript.

Reviewer 3 Report

I have only one note:

Page no.12, line 306 have to be p0 and the same page in line 312

Author Response

We have made the change in both lines and in the others that this parameter appears.